# Topic Extraction and Interactive Knowledge Graphs for Learning Resources

Ahmed Badawy [1,2,*], Jesus A. Fisteus [1], Tarek M. Mahmoud [2,3] and Tarek Abd El-Hafeez [2,4]

1   Telematic Engineering Department, University Carlos III de Madrid, 28911 Legane, Spain;
    jesus.arias@uc3m.es
2   Computer Science Department, Faculty of Science, Minia University, EL-Minia 61519, Egypt;
    d.tarek@mu.edu.eg (T.M.M.); tarek@mu.edu.eg (T.A.E.-H.)
3   Faculty of Computers and Artificial Intelligence, Sadat City University, Sadat City 32897, Egypt
4   Computer Science Unit, Deraya University, EL-Minia 61765, Egypt
*   Correspondence: ahmed.badawy2000@mu.edu.eg

**Abstract:** Humanity development through education is an important method of sustainable development. This guarantees community development at present time without any negative effects in the future and also provides prosperity for future generations. E-learning is a natural development of the educational tools in this era and current circumstances. Thanks to the rapid development of computer sciences and telecommunication technologies, this has evolved impressively. In spite of facilitating the educational process, this development has also provided a massive amount of learning resources, which makes the task of searching and extracting useful learning resources difficult. Therefore, new tools need to be advanced to facilitate this development. In this paper we present a new algorithm that has the ability to extract the main topics from textual learning resources, link related resources and generate interactive dynamic knowledge graphs. This algorithm accurately and efficiently accomplishes those tasks no matter how big or small the texts are. We used Wikipedia Miner, TextRank, and Gensim within our algorithm. Our algorithm's accuracy was evaluated against Gensim, largely improving its accuracy. This could be a step towards strengthening self-learning and supporting the sustainable development of communities, and more broadly of humanity, across different generations.

**Keywords:** education; e-learning; topic identification; natural language processing; interactive knowledge graph

## 1. Introduction

The sustainable development of the community is a great responsibility and a lofty goal that could be achieved in many ways; education is a very important one of them. Education quality is the main indicator of a community's development [1], however, the cost of a high quality education is unaffordable in many countries [2], especially with the effect that COVID-19 has had on many systems [3]. In this context, e-learning was an inevitable solution: it can be used by educational systems as an affordable, safe, and progressing alternative to classical education. Although e-learning is not new, the current circumstances require more capabilities and efficiency from e-learning systems. One of the many advantages of e-learning systems for their users is that it can be available anytime and anywhere, with the lowest costs. Learning resources are available in a wide range of formats that cover almost anything that one could need to learn [4]. E-learning is also a great facilitator of self-learning, as it helps learners to autonomously get appropriate materials for the skills they want to develop, without the need to take full courses at formal education institutions such as schools and colleges. Self-learning has the potential to change the way people are going to be educated. As every student is unique, the education system has the opportunity to adopt new tools that help it to adapt uniquely, according to every

individual student's needs. As reported in [5], 91% of elementary school students and 71% of middle and high school students use e-learning for learning things on their own.

So far, we can note that students in the current era are lucky to have all these possibilities, tools and educational resources at their fingertips. Moreover, several studies have shown that the number of available learning resources is growing every day [6,7].

However, the growing number of learning resources is a double-edged sword, as it will provide a great opportunity for students to learn anything they wish in different styles and ways of exploration, however, it also represents a challenge for students to find and select the required one among this growing ocean of resources.

Therefore, learners would benefit from having new tools and techniques for exploring, browsing and searching learning resources. Those tools would not only save learners time and effort, but also protect them from boredom.

The massive progress of technologies such as Artificial Intelligence, Machine Learning [8], Natural Language Processing [9], Semantic Web [10] and Virtual Reality [11] could facilitate building such tools.

We plan to apply results in these areas to build an e-learning system, inspired by the Learning as a Network learning theory [12], which aims to facilitate the learning process for students in different ways: (1) a topic extraction system, to automatically obtain the main topics from learning resources and build a multilayer topic structure; (2) a clustering system, to connect learning resources based on the relationships between their topics; (3) dynamic interactive knowledge graphs, to show learning resources in an easy and searchable dynamic connected graph; and (4) a social learning network, to connect different users with common interests or common learning goals, and to facilitate self-learning. In particular, in this paper we focus on its first three features: topic extraction, clustering and dynamic interactive knowledge graphs, leaving the social learning network for future work. Our main contribution is a topic extraction algorithm that is able to analyze the text of learning resources, including large books, and identify their main topics as unambiguous references to categories in the Wikipedia collaborative encyclopedia. Based on these extracted topics, the system is able to connect and group related learning resources. These connections can be presented to learners in the form of knowledge graphs that they can browse graphically and interactively in order to search and explore resources they might be interested in.

The rest of this article is organized as follows. Section 2 reviews related research. Section 3 describes the research methodology we followed and our proposal, including a thorough description of our topic extraction algorithm and some examples of the knowledge graphs produced by the system. Section 4 describes the experiments we performed on the topic extraction algorithm and their results, which are discussed and compared to research work in Section 5. Finally, Section 6 concludes the article and describes our future lines of work.

## 2. Literature Review

Automatic Topic Identification in the current information era is vital for finding the required learning material among the daily giant growth of data on the Web. To establish unequivocal topic identifiers we need a comprehensive and up-to-date vocabulary, knowledge source, or ontology that could be used for representing any possible topic a learning resource might be about. The online encyclopedia Wikipedia is the most feasible choice for this task. As such, it represents a giant multilingual database of concepts and semantic relations. A specific topic of an article can be identified with the identifier of the Wikipedia article that describes that topic. Wikipedia Miner [13] is an open-source software system that uses the articles in Wikipedia to extract and identify the main topics in text through its rich semantics, so we use it in our system. It applies parallelized dump processing and machine-learned semantic relatedness measures. Furthermore, it contains a topic detector feature that gathers all labels in the document. For most documents, Wikipedia probably knows something about the topics discussed and could likely add additional information.

Once Wikipedia Miner detects the topics within a document, it is easier for automatic systems to process those learning resources for tasks such as classification, recommendation, or data retrieval.

Wikipedia and Wikipedia Miner have been used in many fields such as automatic topic indexing [14], document clustering [15], document summarization [16], the classification of multilingual biomedical documents [17], converting concept-based representations of documents from one language to another [18], identifying the prerequisite relationships among learning objects [19], classifying news articles [20], evaluating and classifying Open Educational Resources (OERs) and OpenCourseware (OCW) based on quality criteria [21], and for group recommendation by combining topic identification and social networks [22].

As we can see, Wikipedia and its semantic information can be used for a wide variety of applications and fields due to its reliability, efficiency, and the continuous updating of content by a large community of users.

The process applied by Wikipedia Miner begins by gathering overlapping word $n$-grams (where $n$ ranges between one and the maximum label length in Wikipedia) from the processed document and consulting the label vocabulary to ascertain which terms and phrases correspond to concepts in Wikipedia. A limitation of this technique is that the bigger the text size in the document, the harder the process will be.

We also used TextRank [23], which is a graph-based ranking model for text processing. TextRank is an unsupervised algorithm for the automated summarization of text. The algorithm can obtain the most important keywords in a document without the need for a training corpus or labeling. This algorithm was later improved upon by Federico et al. in [24], by introducing something called a "BM25 ranking function", which is a ranking function widely used in the state of the art information retrieval tasks. BM25 is a variation of the TF-IDF model using a probabilistic model. They achieved an improvement of 2.92% above the original TextRank result by using BM25. The combination of TextRank with modern information retrieval ranking functions such as BM25 and BM25+ creates a robust method for automatic summarization that performs better than previous standard techniques [24].

Next, we are going to discuss topic identification techniques and some of their applications. Topic identification is a crucial field in natural language processing. This technique has many applications such as information retrieval, document summarization, topic detection and tracking, text classification, etc.

In [25] Chris et al. presented TopCat (Topic Categories), a technique for identifying topics that recur in articles in a text corpus. Chris et al. identified related items based on traditional data mining techniques. Frequent item sets are generated from the groups of items, followed by clusters formed with a hyper graph partitioning scheme. Natural language technology was used to extract named entities from a document, and then look for frequent item sets. Next, groups of named entities were clustered, capturing closely related entities that may not actually occur in the same document. Finally, a refined set of clusters was produced, with each cluster representing a set of named entities that refer to a topic.

In [26] Veselin et al. presented an algorithm for opinion topic identification through developing a methodology for the manual annotation of opinion topics with the use of fine-grained subjectivity analysis, which could be useful for question answering, summarization, and information extraction.

In [27] Kino et al. presented a method for automatic topic identification using an encyclopedic graph derived from Wikipedia. Kino et al. used the unsupervised system Wikify [28] to identify the important encyclopedic concepts in an input text automatically. As Kino et al. mentioned "topic identification goes beyond keyword extraction, instead has to be obtained from some repositories of external knowledge". Kino et al. aimed to find topics (or categories) that are relevant to the document at hand, which can be used to enrich the content of the document with relevant external knowledge.

In [29] Freidrich et al. presented a framework for the identification of primary research topics from within a corpus of related publications. Their method uses an unsupervised topic modeling approach to classify new and emerging topics from the entire corpus. Machine learning techniques were used, such as Non-negative Matrix Factorization for Natural Language Processing, as well as an adaptive topic model Bayesian classifier that allows for the identification of new primary topics as papers are added.

In [30] Khader et al. designed an ensemble method for automatic topic extraction from a collection of scientific publications based on a multi-verse optimizer algorithm as the clustering algorithm.

Documents that refer to a specific topic might also refer to several sub-topics of it. Then the single-layer method could present problems to define the intended topics, even when a document is analyzed manually by human experts. A new multilayer topic structure was proposed in [31]. The multilayer topic structure aims to automatically build a multilayer topic structure. This technique is intended to identify sub-topics where units within the same sub-topic should be very similar and units from different sub-topics should be dissimilar. The mentioned paper used the hierarchical agglomerative clustering algorithm to establish the hierarchical topic tree. It started by using each unit as an independent cluster. Then, the similar ones merged together. This process was repeated until all clusters merged into one cluster, and from this hierarchy a tree structure was established.

Our system also has the ability to create this tree structure of topics. We will elaborate on the steps for this extraction in our algorithm in Section 3.

Books with common topics are semantically similar, and could be linked together to provide a network of resources that are related to a general topic. We can provide this information in a Dynamic Interactive Knowledge Graph, which can be updated continuously with every new book added to the database, and can be browsed by every user. This graph-based learning technique is an emerging search field and proved to be effective and useful for the learning process for students and teachers: as Weber et al. mentioned in [32] the graph-based learning technique improves search ability for new sources and provides explainable results and result recommendations.

In [33] Zhijun et al. used an interactive group knowledge graph to visualize the relationship between knowledge points concluding that interactive group knowledge graphs have a significant promotional effect on teachers' online learning, that it is beneficial to teachers' professional development, and that it is an effective method of enhancing the depth and breadth of the interaction of students' learning. In addition, Zhijun et al. defend that the process of generating collective knowledge graphs can enhance students' enthusiasm for autonomous learning and promote meaningful and in-depth interactions between different students.

In order to put our research into context we must also undergo a quick overview on learning theories. In [34] Phillips et al. mentioned Plato's theory, which postulates that individual's knowledge is present at birth, which is called the Theory of Recollection or Platonic epistemology. In [35] Brain et al. mentioned Locke's theory, which postulates that humans are born into the world with no innate knowledge and are ready to be written on and influenced by the environment. There are many other learning theories such as Behaviorism [34], Social learning theory [36], Cognitivism or Gestalt theory [37], Constructivism [38] or Transformative learning theory [39].

Our proposal is based on the Learning as a Network learning theory [12]. This starts with the learner and views learning as the continuous creation of a personal knowledge network. The Learning as a Network builds upon connectivism [40,41]. Connectivism is a recent theory of networked learning [42], which focuses on learning as making connections. The Learning as a Network theory is considered the most appropriate for the current conditions and for the tools that we will study and use in this research, such as the interactive knowledge graph that we will produce and illustrate in Section 3.

## 3. Materials and Methods

As was explained in Section 1, the main contribution of our work is the algorithm we designed to extract main topics from textual learning resources. We will describe the algorithm in this section, as well as how its results can be applied to clustering learning resources and creating interactive and dynamic knowledge graphs.

We applied a quantitative research methodology to evaluate our proposal. The experiments we performed on the algorithm are described in Section 4. Their objectives were to optimize a configuration parameter of the algorithm (the parent category level, which will be explained later in this section), to evaluate the precision of the main topics extracted by the algorithm and measure the time it takes to produce those topics, in comparison to another state of the art algorithm. The experiments were based on a dataset of more than 500 full text learning books we collected from several repositories. We describe this dataset in Section 4.

Our system uses Semantic information [43] to store the information about every student and every learning resource, as it is machine-understandable, productive in inference and recommendation, and effective in social learning networks. For students, we record their age, learning goals, current and required skills, and their learning background. For learning resources, we record title, author, abstract, level, type, and size. In this paper we will focus on analyzing textbooks to identify main topics and classify them to produce dynamic interactive graphs of learning resources.

The learning resources analysis phase starts by extracting text from books, then processing text and extracting main keywords using natural language processing techniques. As we mentioned before, we are going to use TextRank [23] combined with the BM25 ranking function [24] to improve the results. By using Gensim (we used Gensim 4.1.2 https://pypi.org/project/gensim/ (accessed on 29 October 2021)), which is a very popular open-source library for unsupervised learning implemented in Python [44], we can apply that algorithm on the text of the books in order to extract main keywords.

To identify the Main Topics in each learning resource we have to understand the difference between Keywords and Main Topics. Keywords are important words in a text that could represent a topic mentioned in the book. They are an exact match to text that appears in the book. Main Topics are the most important topics found in the book according to the category retrieved from the Wikipedia database, which have a specific id and a URL (https://en.wikipedia.org/wiki/Wikipedia:FAQ/Categories (accessed on 29 October 2021)). They are not necessarily an exact match to words mentioned in the book, since they could also appear with semantically related words.

An example of a keyword is the word "business" appearing in the text of a book. A Main Topic, related to that keyword, could be the Wikipedia category titled "Business", with identifier 771152 and URL https://en.wikipedia.org/wiki/Category:Business (accessed on 29 October 2021).

Therefore, Main Topics consist of a category in Wikipedia, which could provide more information and link to related resources that are not necessarily mentioned in the original text.

All the steps of our algorithm are explained in the next Algorithm 1 pseudo code and Figure 1.

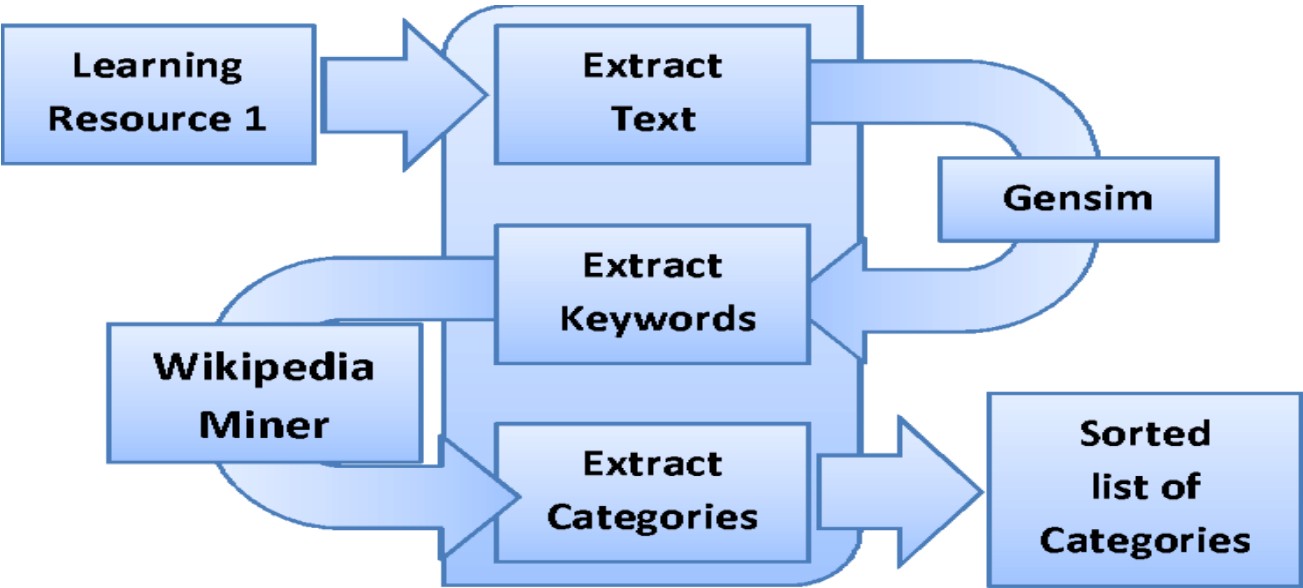

**Figure 1.** Our algorithm structure.

The Main Topics extraction phase uses Wikipedia Miner [13] where Wikipedia's articles, categories and redirects are represented as classes, and can be efficiently searched, browsed, and iterated over. This is a platform for sharing data mining techniques based on Wikipedia data. This data is gathered from Wikipedia's XML dumps. We used a dump of the English version of Wikipedia dated on 22 July 2011, which includes 3.3 million articles and 37.4 GB of uncompressed markup. Wikipedia Miner is the fundamental tool in our Topic Identification system that enables us to assign the right main topic to the learning resources. There is also a parent category feature that enables us to find the parent category of the specified category/topic.

In our initial tests we ran Gensim with the Wikify service [28] of Wikipedia Miner, which is used for keyword extraction and word sense disambiguation, however we found many missing results, even after repeating the experiments several times. To solve this problem we augmented the combination of Gensim and Wikipedia Miner (Wikify) with the Search service of Wikipedia Miner, which solved the problem by searching all available articles for those that are related to the identified keywords. The actual data of this experiment will be presented in the explanation of experiment 2 in Section 4.

For each learning resource, after extracting the main keywords with Gensim, we use Wikipedia Miner to extract main topics, then for each topic we identify the parent categories at different levels. We iterate over the retrieved categories to extract multi-level results. This step produces a hierarchy of topics that represents a multilayer topic structure as mentioned in [31]. When we mention in this paper the second level of the parent category we mean the second iteration of the parent category service, and so on.

---

**Algorithm 1** Pseudo code

---

Here we explain our Algorithm for Topic Identification:

1.　For each (Learning Resource) in the dataset:
2.　Extract (Text)
3.　Process (Text) with Gensim to extract (Keywords)
4.　"Process (Keywords) with Wikipedia Miner services (Wikify, Search) to extract (Article IDs)"
5.　Copy (Keywords) to (Wikify_Keywords)
6.　Copy (Keywords) to (Search_Keywords)
7.　While (number of retrieved Article IDs < 20)
8.　Process (Wikify_Keywords) with (Wikify) to extract topic IDs
9.　Remove the last keyword from (Wikify_Keywords) list "to enable Wikify to find more results"
10.　If length of (Wikify_Keywords) = 0:
11.　For each Keyword in (Search_Keywords):
12.　Process Keyword with (Search) to extract topic IDs
13.　"Process (Article IDs) with Wikipedia Miner services (exploreArticle, suggest, exploreCategory,
14.　parentCategory) to extract (Definition, Title)"
15.　Append all (Article IDs) in one Article IDs list
16.　For each ID in Article IDs:
17.　Process ID with (exploreArticle) to extract Definition and Title
18.　Process ID with (suggest) to extract suggested Categories IDs
19.　For each suggested Category ID:
20.　Process ID with (exploreCategory) to extract Definition and Title
21.　Get Parent Category with (parentCategory) property
22.　"[(parentCategory) could be repeated to reach the required level
23.　for (2nd–3rd–4th) level of parent categories"
24.　Collect all Parent Categories with repetition numbers.
25.　Sort Parent Categories from high to low repetition to extract the most important ones.

---

Now we will show some examples of our algorithm's results in Table 1.

**Table 1.** Examples of our algorithm's results.

| Book main topic (Biology) |
|---|
| 1st level:<br>(Immune system/Organ systems/Microbiology/Biology/Clinical pathology)<br>2nd level:<br>(Biology/Life/Natural sciences/Botany/Anatomy).<br>3rd level:<br>(Biology/Life/Society/Natural sciences/scientific disciplines).<br>4th level:<br>(Life/Biology/Nature/Natural sciences/Universe). |
| Book main topic (Math) |
| 1st level:<br>(Mathematical analysis/Calculus/Subdivisions of mathematics/Analysis/Integral calculus)<br>2nd level:<br>(Subdivisions of mathematics/Mathematics/Abstraction/Dimension/Geometry)<br>3rd level:<br>(Mathematics/Subdivisions of mathematics/Abstraction/Structure/Dimension)<br>4th level:<br>(Structure/Scientific disciplines/Academic disciplines/Abstraction/Dimension) |

As we can see from the examples, the higher the level the more general and comprehensive the results are, yet at some level those results are too broad and not so useful. We describe in the Results section how we empirically chose the best level of parent category to use in our system.

For each learning resource we select the top five main topics retrieved by the algorithm. After assigning those main topics to the learning resource we will establish the base for the dynamic interactive knowledge graph that will contain all the learning resources. We insert each learning resource and main topics as nodes, then we create a link between each main topic and all related learning resources. The result is a connected graph of related learning objects as shown in Figures 2–5.

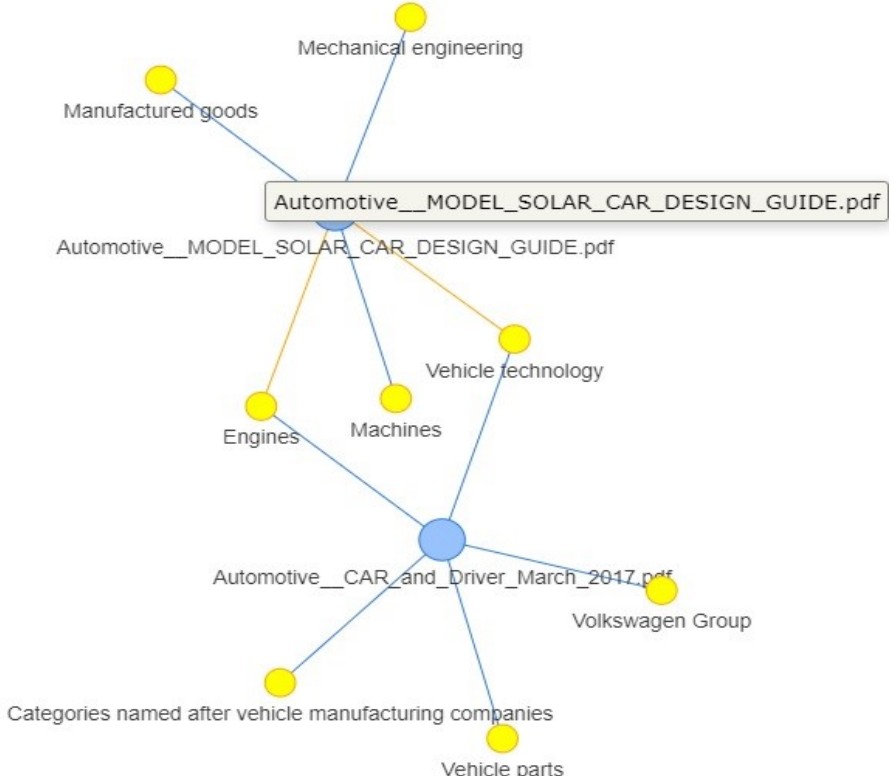

**Figure 2.** Relation between learning resource (blue), and main topics (yellow).

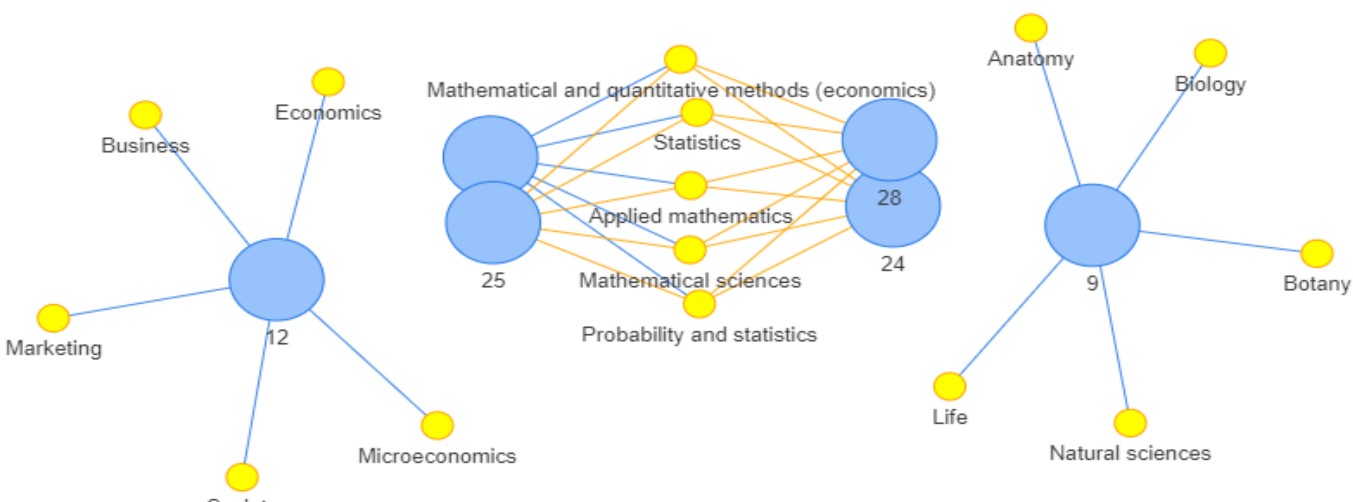

**Figure 3.** Relation between learning resources (blue), and main topics (yellow).

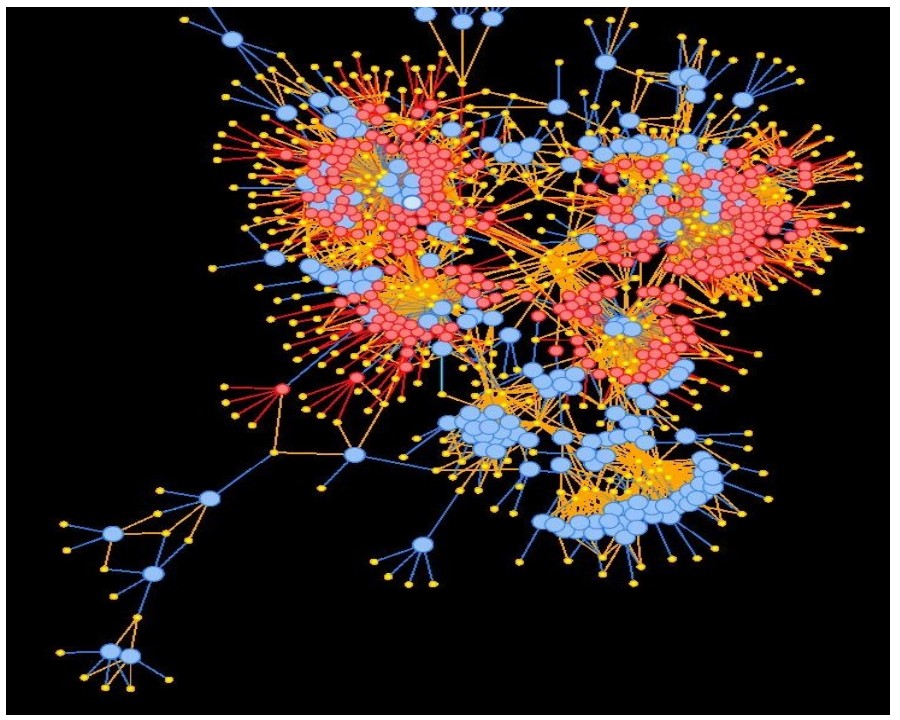

**Figure 4.** Relation between one learning resource (blue), main topics (red), and chapters (yellow).

**Figure 5.** Relation between learning resources (blue), main topics (yellow), and chapters (red).

Students can use this graph in different useful ways. They can find the required main topic and browse all related learning objects, find a learning object then identify all related main topics, or find related learning objects and explore common main topics between them.

Building knowledge graphs from the text in learning resources is another key feature in our platform, where we can focus on self-directed learning to enable students to identify the required topic or learning resource easily and in a joyful way. In Figure 2 we can see the relation between learning resources (in blue), and main topics (in yellow).

In Figure 3 we can see the relation between learning resources (in blue), and main topics (in yellow).

In Figure 4 we can see the relation between one learning resource (in blue), main topics (in red), and chapters (in yellow). To obtain this graph we apply our algorithm separately on each learning resource and each chapter on it.

In Figure 5 we can see the relation between learning resources (in blue), main topics (in yellow), and chapters (in red).

As was explained in Section 1, we plan to complete in the future our e-learning system with a social learning network. This system would be responsible for storing students' information including their learning goals, current and required skills, and their learning background. The system will assign to each student's learning path every learning resource they finish. With this information and the topics our algorithm extracts, the system will be able to recommend learning resources or complete learning paths for other students with the same goals or learning background.

After we have explained our methodology and the structure of our system, we are going to illustrate our experiments to evaluate our topic identification algorithm.

## 4. Results

We conducted two experiments to evaluate our algorithm. In the first experiment we ran our algorithm on a dataset of 579 books with different configurations of parent category level, in order to analyze its accuracy and select the optimal value for that parameter. In the second experiment we compared the accuracy of our algorithm's results against another topic extraction technique, and we also compared computation time for both of them.

These experiments were run on a collection of 579 books in different subjects, levels, and from different learning repositories. We selected them for the levels of secondary school and first years of university/college. The topics of the books included: Anatomy, Art, Astronomy, Automotive, Biology, Biostatistics, Business, Chemistry, Computational Physics, Computing, Economics, Education, Engineering, First Aid, Hardware, History, Humanities, Languages, Law, Math, Medicine, Organic Chemistry, Parenting, Physics, Physics and Environment, Psychology, Science, Social sciences, Software, Statistics, and Travel.

### 4.1. Experiment 1

We extracted the results from the system in four different levels (1st, 2nd, 3rd, and 4th) depending on the number of parent categories used in Wikipedia Miner. We used top-5 accuracy [45] to evaluate the results in each level to identify the most suitable level to use.

For each learning resource we retrieved the top five main topics, and then we evaluated each topic manually and gave each learning resource a score out of five. If the resource got five of five then all the five topics were correct, however if the resource got two of five, then only two of the topics were correct.

Next we summed all the results and divided the total by the number of resources to find the average score (average true value).

Figure 6 shows, for each parent category level, the number of resources for which the algorithm got the five topics correct, four topics correct, etc.

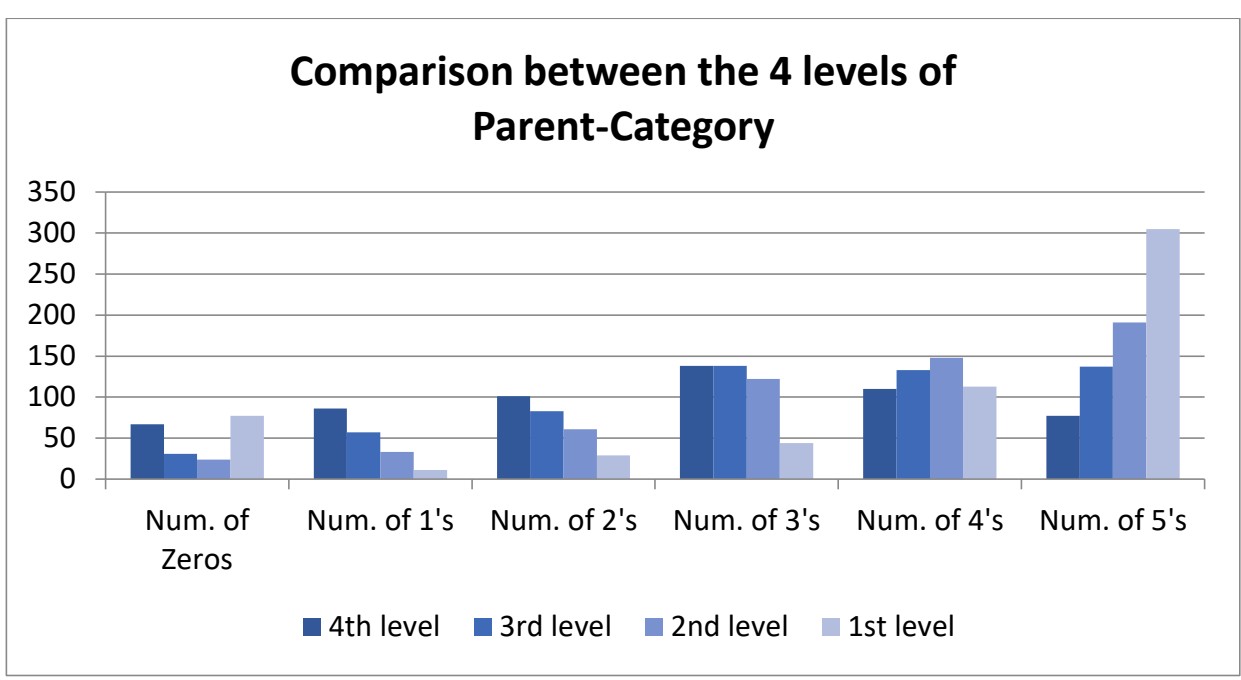

**Figure 6.** Comparison between the four levels of parent-category.

Table 2 lists, for each parent category level, the percentage of documents for which at least one of the five topics was correct ("True topic identified"), the percentage of documents for which none of the five topics was correct ("False topic identified") and the average number of correctly identified topics ("Average true").

**Table 2.** Comparison between the four levels.

|  | True Topic Identified | False Topic Identified | Average True |
|---|---|---|---|
| 4th level Parent-Category | 0.884 | 0.116 | 0.527 |
| 3rd level Parent-Category | 0.946 | 0.054 | 0.640 |
| 2nd level Parent-Category | 0.959 | 0.041 | 0.714 |
| 1st level Parent-Category | 0.867 | 0.133 | 0.752 |

We can find that the results of using both the first and second level parent categories show a high number of resources with a five of five in top-5 accuracy, especially for the first level. In addition, the average number of correctly identified topics is higher (0.752) for the first level than for the second level. Therefore, we chose the first level of the parent category in our algorithm for the rest of the experiments, although we retain the ability to use many levels to produce hierarchical structure of main topics.

*4.2. Experiment 2*

The goal of this experiment was to compare the computation time and the accuracy of our algorithm against other topic identification techniques. As Gensim and Wikipedia Miner have the ability to identify topics from text, we compared them against our algorithm, which is built on top of Gensim and Wikipedia Miner as was explained in the Methodology section.

In this experiment we sampled 350 books from the dataset that we used in the first experiment. The books in this sample of the dataset contain between 95 and 1800 pages. In Figure 7 we can see the distribution of the number of pages of the books in the dataset.

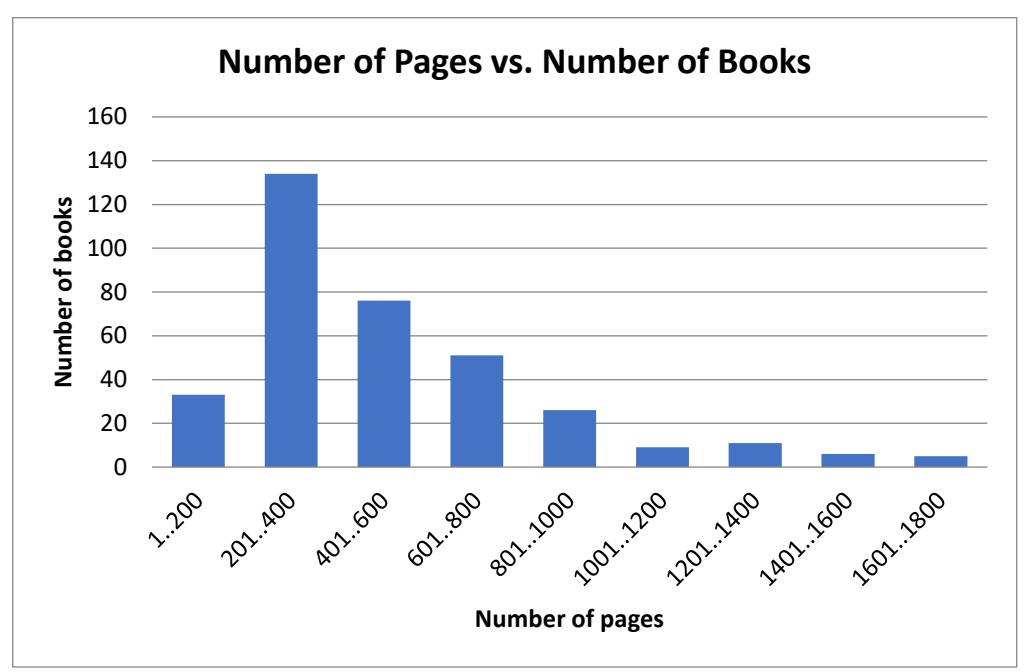

**Figure 7.** Number of pages vs. number of books.

However, we have seen that computation time correlates better with text size (in terms of number of characters) than with the number of pages of a book. That happens, for example, as we are processing text, and some books contain a big number of pages filled with charts, figures, and pictures, which are not processed by the algorithms. The histogram of the number of characters per book is shown in Figure 8.

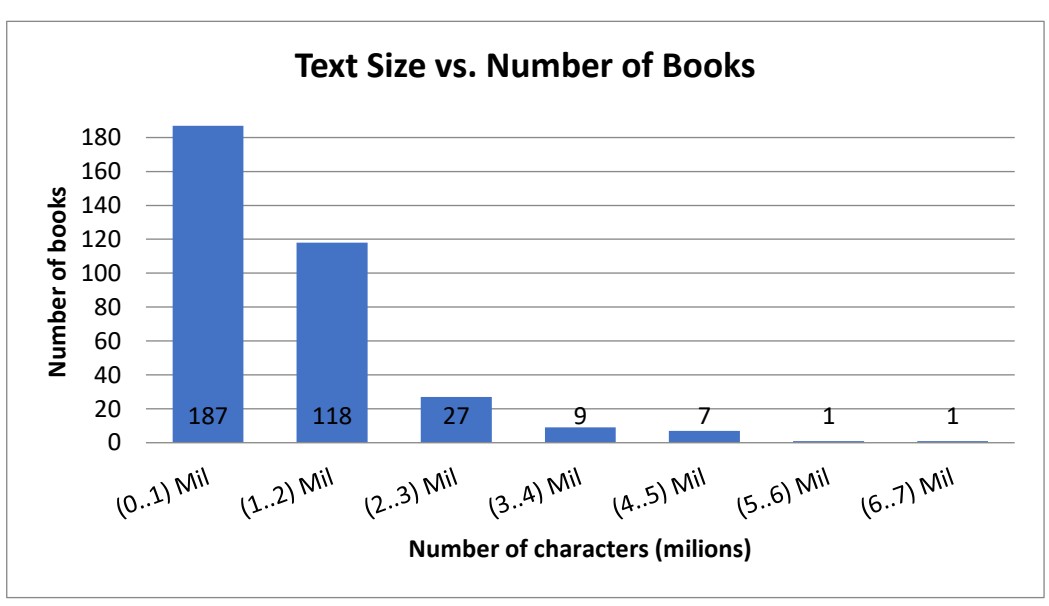

**Figure 8.** Text size vs. number of books.

When we started our comparison we found that the maximum number of pages that we can process with Wikipedia Miner is between 25 and 35 pages. With more pages the

Wikipedia Miner server frequently stopped responding. On the other hand, Gensim had no problem with the number of pages, so we continued our experiment comparing Gensim alone against our algorithm.

A first version of our algorithm was only based on Gensim and the "Wikify" Wikipedia Miner service. However, as we mentioned in the Methodology section, it did not produce any topic suggestion for a big portion of the books in the dataset, more than 200 books out of 350. As a result of that, the final version of our algorithm uses, in addition, the "Search" Wikipedia Miner service for the cases in which the "Wikify" service does not produce any result. The final version of the algorithm produced suggestions for all but 11 books in the dataset.

First, we compared computation time for both Gensim and our algorithm, taking text size into account, in Figures 9 and 10. Two plots are presented: one that presents a data point for each document and another one that groups books by ranges and shows the average computation time for the books at each range.

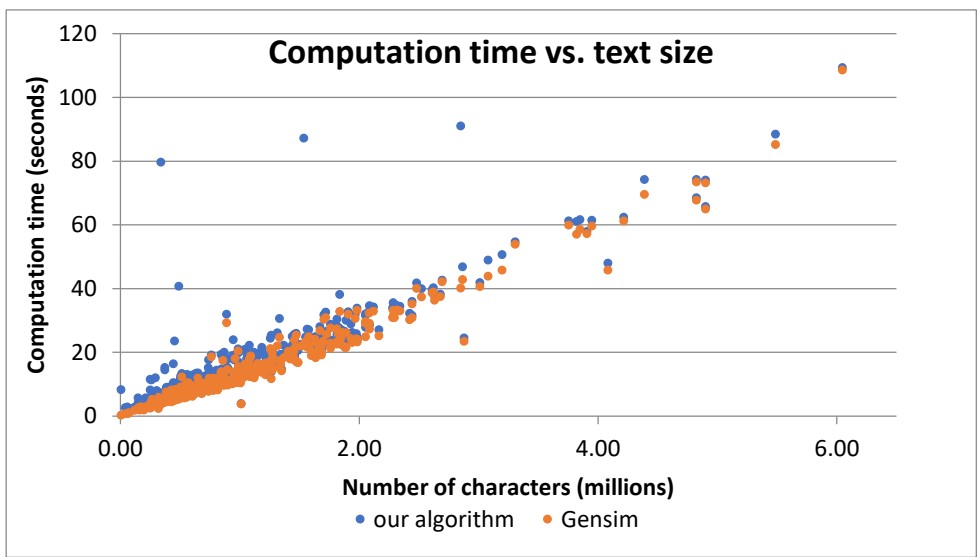

**Figure 9.** Computation time vs. text size.

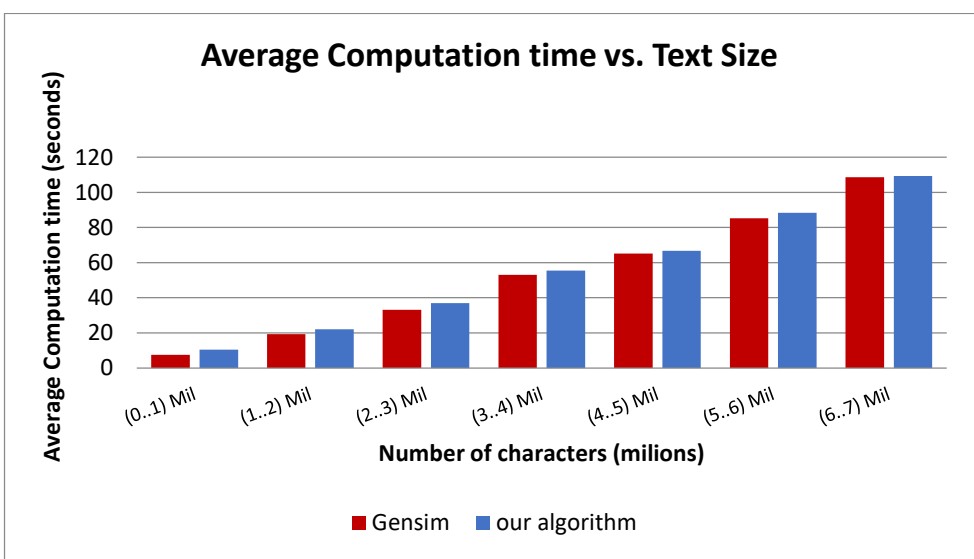

**Figure 10.** Average computation time vs. text size.

Figures 11 and 12 show the same data in the form of box plots, where minimum, maximum and median values for each range, as well as their 25 and 75 percentiles, can be identified.

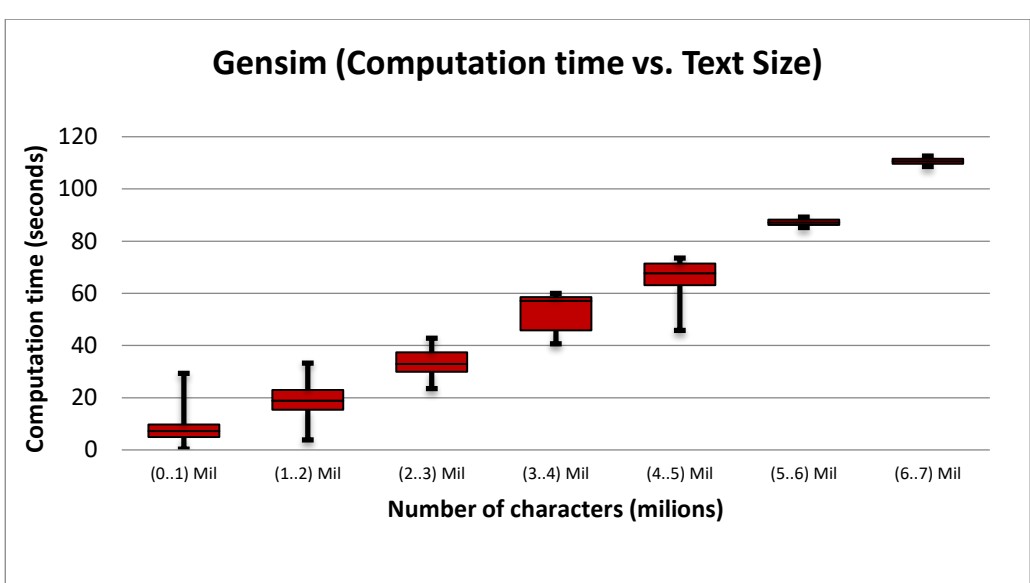

**Figure 11.** Gensim (computation time vs. text size).

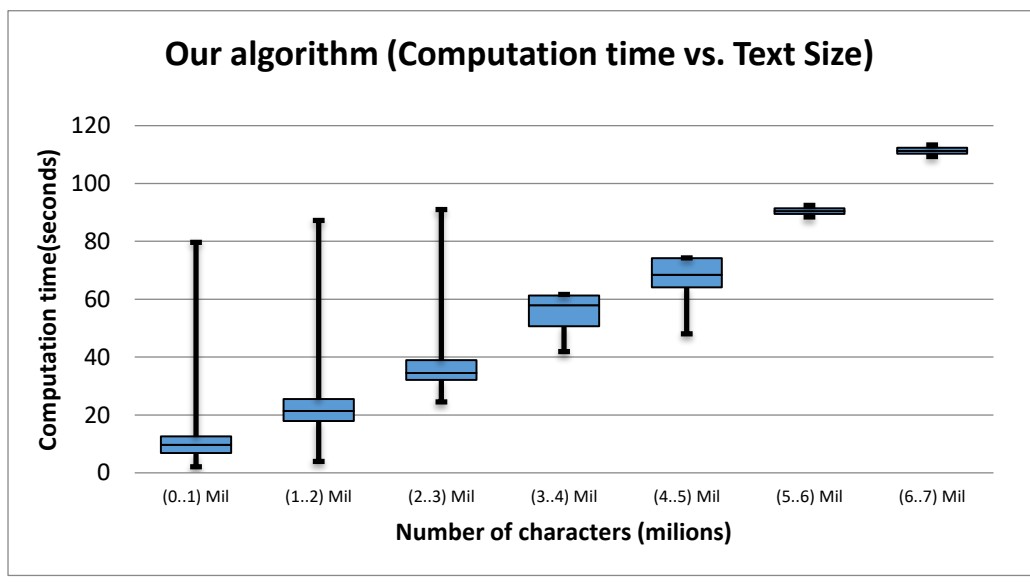

**Figure 12.** Our algorithm (computation time vs. text size).

Table 3 presents the total computation time for the 350 books in the dataset and the average computation time per book.

**Table 3.** Total computation time.

|  | Total Time (s) | Average Time (s) |
|---|---|---|
| Gensim | 5721 | 16.35 |
| Our algorithm | 6739 | 19.25 |

We can see in the plots that our algorithm takes slightly more time than Gensim, which was expected as our algorithm runs Gensim internally. The overhead is due to the use of

Wikipedia Miner services on the results provided by Gensim, including communication time caused by the running of a different server. In spite of these extra tasks, the average overhead of our algorithm is 17.7%, lower than 3 s per book.

After having analyzed computation times and their relation to text size, we will analyze the accuracy of the results provided by Gensim and our algorithm. First, we present an example of the results that we got for three books we sampled from the dataset in Table 4.

**Table 4.** Example of results.

| Book Information | Title: Principles of Accounting, Volume 1: Financial Accounting Pages: 1055 Category: Business | Title: College Physics for AP® Courses Pages: 1694 Category: Physics | Title: Anatomy & Physiology Pages: 1420 Category: Anatomy |
|---|---|---|---|
| Wikipedia miner | Fails to provide an answer. | Fails to provide an answer. | Fails to provide an answer. |
| Gensim | Companies/company/ accounting/accountants/ account/accounts/ accountant/accountable/ accountancy/accounted | Credit/credited/credits/ figures/figure/figured/ figuring/energy/energies/force | Figure/figures/figurative/ cell/cells/called/calling/ muscle/muscles/blood |
| Our Algorithm | Accountancy/Investment/ Financial markets/ Accounting systems/ Management/ Applied sciences/ Business/ Business economics/ Accountability/ Legal entities | Physics/ Fundamental physics/ concepts/ Concepts by field/ Introductory physics/ Physics education/Motion/ Space Time/Force/Phenomena | Anatomy/Biology/ Tissues/ Organs/ Musculoskeletal system/ Subjects taught in medical school/ Cardiovascular system/ Circulatory system/ Medical specialties/ Greek loanwords |

We can see from the results that our algorithm provides the most relevant and accurate results. In addition, whereas Gensim produces just words, our algorithm produces unambiguous Wikipedia categories, referenced by their unique identifier at Wikipedia. The identifiers are not shown in the table for readability.

In order to evaluate accuracy, in this experiment we made Gensim and our algorithm produce 10 topics for each book.

For each book we evaluated these 10 results and computed accuracy (number of correct topics divided by 10). Topic correctness was decided by a human evaluator. We can see in Table 5 that our system achieves a much better accuracy than using Gensim alone.

**Table 5.** Top-10 accuracy.

| | Our Algorithm | Gensim |
|---|---|---|
| Average accuracy | 0.712285714 | 0.240857143 |

Then we evaluated these topics by using top-k accuracy [45]. We compared top-1, top-5, and top-10 accuracy in Figure 13 and in Table 6. For each book we evaluated the results according to the three levels. If the first topic returned by the algorithms is the main category of the book or close enough to it, then the book counts for top-1 accuracy. If one of the first 5 topics is the main category or close enough, then the book counts for top-5 accuracy. Likewise, if one of the first 10 topics is the main category or close enough, the book counts for top-10 accuracy. Again, a human evaluator decided that book by book.

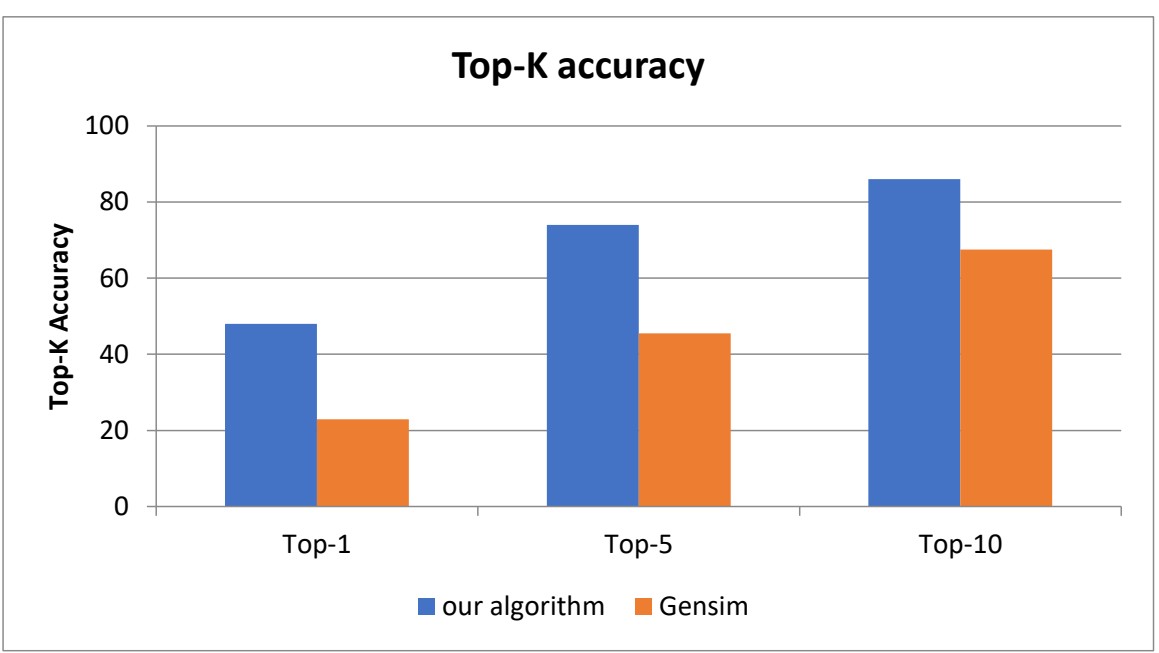

**Figure 13.** Top-K accuracy.

**Table 6.** Top-k accuracy.

|  | TOP-1 | TOP-5 | TOP-10 |
|---|---|---|---|
| Gensim | 22.9% | 45.5% | 67.5% |
| Our algorithm | 48.0% | 74.0% | 86.0% |

Now we want to analyze the relation between accuracy and text size. We start by plotting all accuracy values for all books against text size in Figure 14.

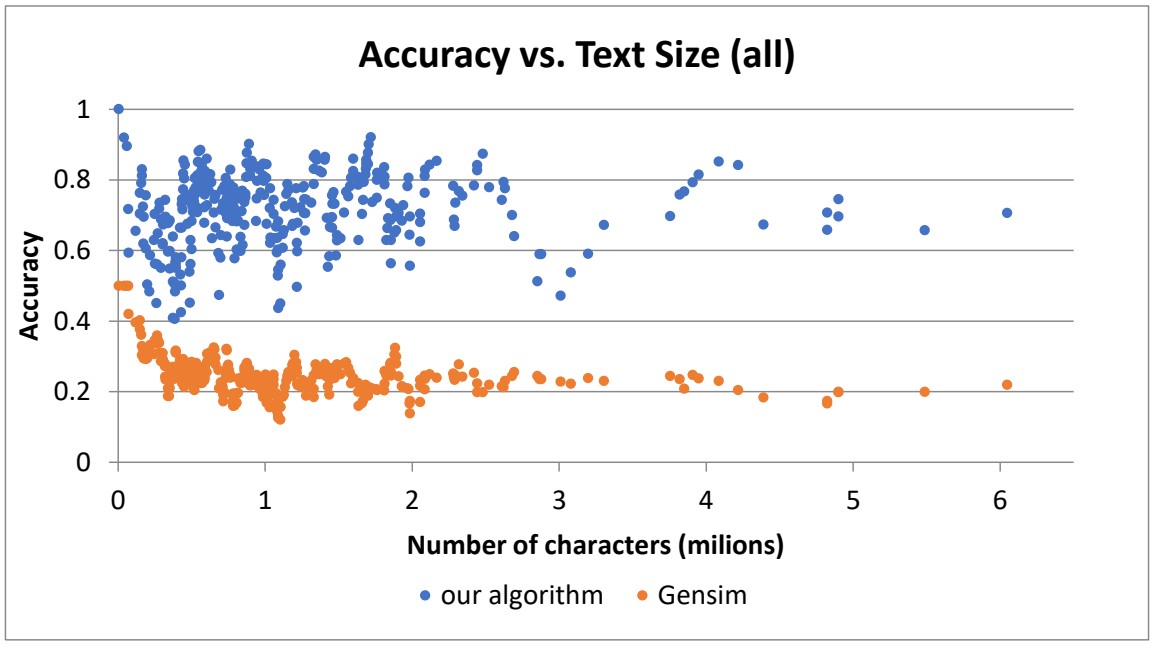

**Figure 14.** Accuracy vs. text size (all).

Next we present top 1, top 5, and top 10 values for Gensim and our algorithms, in groups of different text sizes in Figure 15.

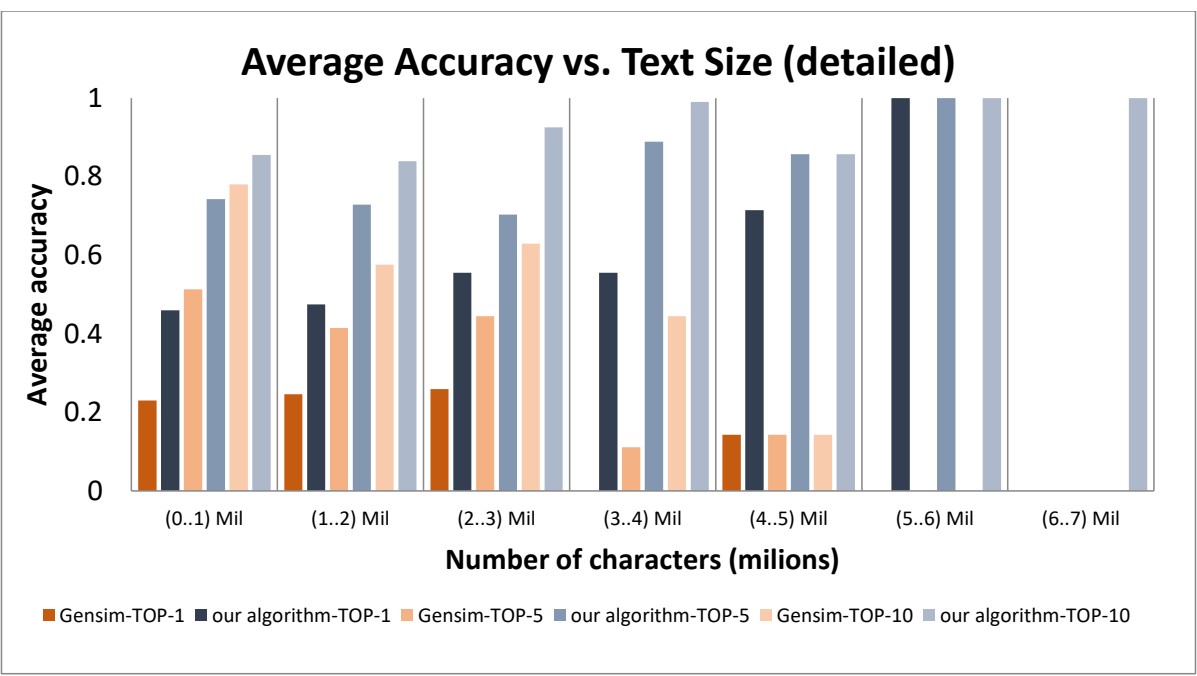

**Figure 15.** Average accuracy vs. text size (detailed).

We can see in this graph that there are 0-sized bars with text sizes between 5 and 7 million characters. The reason is the small number of books these groups contain, as we can see in Figure 8.

Finally, the average values of the accuracy is calculated and compared to groups of text sizes in Figure 16.

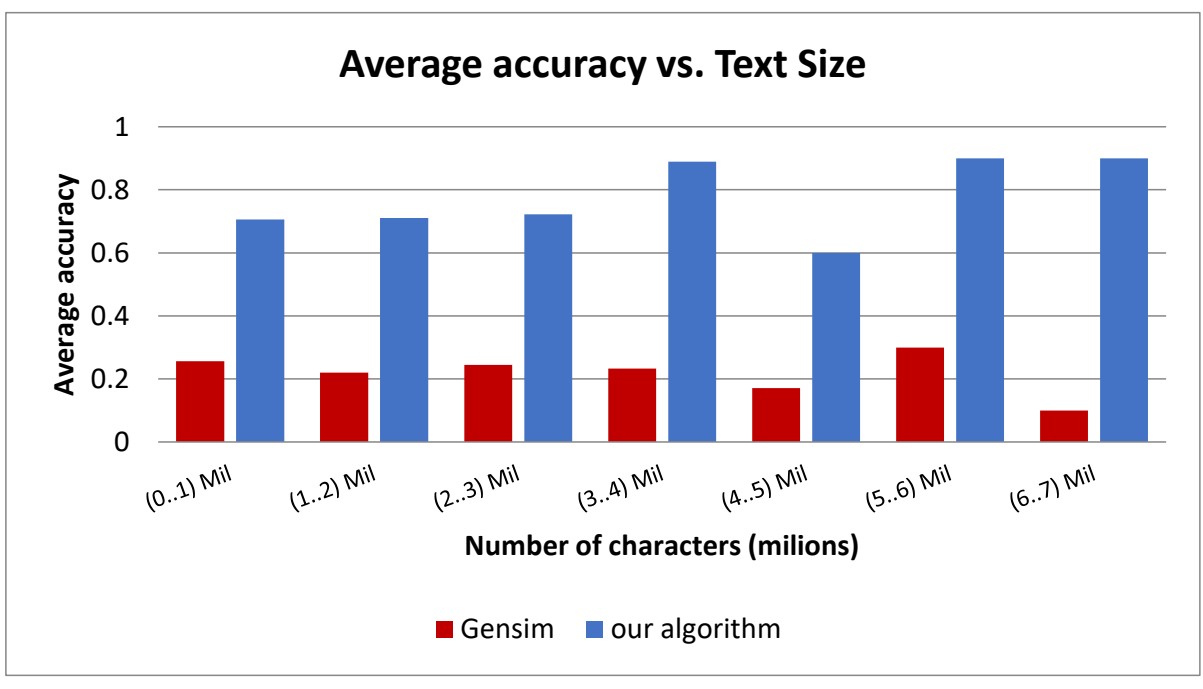

**Figure 16.** Average accuracy vs. text size.

## 5. Discussion

We developed an algorithm to process learning topics in order to extract main topics. In our algorithm we start by extracting text, then we use Gensim to extract the main keywords from the text, and finally we use Wikipedia Miner with both the Wikify and Search services to extract main topics. We can use different levels of the parent category, so we tried many levels in experiment 1 to find out the best one to use in our algorithm.

From the main topics extracted by our algorithm we produced interactive knowledge graphs to facilitate finding the required main topic, browsing all related learning objects, finding a specific learning object, identifying all related main topics, finding related learning objects, and exploring common main topics between learning objects.

In experiment 1 we produced results with our algorithm for different levels of parent category. We found that the first level is the best choice to use in our algorithm, however we can use different levels if a hierarchical topic structure is needed.

In experiment 2 we proved that the Wikify service of Wikipedia Miner does not work properly for the books in our dataset, as it does not produce results for more than 57% of the books. With our algorithm this happens for just 3% of them.

We also analyzed the computation time for both our algorithm and Gensim. As expected, our algorithm takes slightly more time than Gensim, due to it using Wikipedia Miner services on the results provided by Gensim. What is important to mention is that the experiment shows the relationship between the computation time of our algorithm and text size is linear.

Finally, we showed that our algorithm largely outperforms Gensim in terms of the accuracy of the topics it extracts, with the additional advantage of producing them in an unambiguous way as references to Wikipedia categories. According to our experiments, text size does not affect the accuracy of the topics our algorithm extracts.

### 5.1. Comparison to Previous Studies

Several attempts have been made in the past to accomplish some of our goals as we mentioned in Section 2.

In [13] David and Ian presented Wikipedia Miner although they evaluated it for news articles. In [14] Olena et al. used Wikipedia and Wikipedia Miner for automatic topic indexing, however they did not mention the document sizes. In [27] Kino et al. presented a method for automatic topic identification using an encyclopedic graph derived from Wikipedia. They used the unsupervised Wikify service. However, as we report in Section 4, the Wikify service is not able to produce results for larger resources such as many of the books in our dataset.

In [25] Chris et al. presented TopCat, which was effective in identifying topics in collections of news articles. In [26] Veselin and Claire presented an algorithm for opinion topic identification. In [29] Freidrich et al. presented a framework for the identification of primary research topics. In [30] Khader et al. designed an ensemble method for automatic topic extraction from a collection of scientific publications. However, all this research focuses on news articles, opinion topic identification, research topics or scientific publications, which are smaller than the learning resources in our dataset.

We can conclude that the main advantage of our proposal in comparison to previous work is its ability to work on large resources such as big-sized learning books that normally contain hundreds of pages and even reach two thousand pages in some cases.

### 5.2. Research Restrictions and Practical Implications

So far, we have applied our algorithm on textual educational resources only. Recent advances in speech-to-text techniques suggest that it could also probably work for extracting the main topics of other kinds of learning resources such as videos and audio. However, further research is needed in order to prove that.

Applying our algorithm requires the computational power of a high-end computer or a computer cluster, especially for running the Wikipedia Miner services, which work on a

massive data base of information extracted from Wikipedia. Since this task is intended to be run at the core of the e-learning system, this limitation would not affect learners, which would be able to use the system from the device of their choice.

## 6. Conclusions

At the end of our experiments, and after all the facts that we have seen and illustrated, we can conclude that Wikipedia Miner is a very useful tool that can be used in various fields, however, it presents an important limitation with respect to the size of the text it can process as one unit. Gensim is also an effective tool although our algorithm, which works on top of Wikipedia Miner and Gensim, outperforms both of them and fixes their disadvantages.

Our algorithm takes a very small amount of extra computing time yet produces results with a higher value of accuracy. Using our algorithm can also produce dynamic interactive knowledge graphs that could be augmented in any e-learning platform or used on its own for different goals. This gives learners the ability to explore a huge number of learning resources in a short time and in an interactive and exciting way. This is a promising way of exploring learning resources with the current growing data rate on the Internet.

In the future we can integrate other tools to analyze learning resources in different formats like video, audio, and graphics. This could produce a strong system for e-learning that will beat the huge amount of learning resources currently available and help learners find the resources they need. We will also complete the system with social learning network features, to help learners socialize during the learning process.

**Supplementary Materials:** The following supporting information can be downloaded at: The list of dataset book is available online at: https://docs.google.com/spreadsheets/d/1hjEcUft-QnYvv5 SVZSAt-EjJbfDGDUzK/edit?usp=sharing&ouid=111160959031654571802&rtpof=true&sd=true (accessed on 29 October 2021), all figures and tables in the article are available online at: https://drive. google.com/file/d/1uIaZkV202K4aRmS8GnLkPKVOZjmnV2m1/view?usp=sharing (accessed on 29 October 2021).

**Author Contributions:** A.B.: Conceptualization, methodology, software, validation, formal analysis, investigation, resources, data curation, writing—original draft preparation, writing—review and editing, visualization, funding acquisition. J.A.F.: Conceptualization, methodology, validation, formal analysis, investigation resources, data curation, writing—original draft preparation, writing—review and editing, visualization, supervision, project administration, funding acquisition. T.M.M.: Conceptualization, writing—review and editing. T.A.E.-H.: Conceptualization, writing—review and editing. All authors have read and agreed to the published version of the manuscript.

**Funding:** The researcher was partially funded by the Egyptian Ministry of Higher Education and Minia University in the Arab Republic of Egypt. [Joint supervision mission from the fourth year missions (2015–2016) of the seventh five-year plan (2012–2017)].

**Institutional Review Board Statement:** Not applicable.

**Informed Consent Statement:** Not applicable.

**Data Availability Statement:** (Third Party Data) Copyright holders of the books in the dataset are third parties and we cannot share book contents. "Disclaimer": we are sharing only the titles of all the books as Supplementary Material.

**Conflicts of Interest:** The authors declare no conflict of interest.

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
