# Peer review of "Topic Extraction and Interactive Knowledge Graphs for Learning Resources"

_sustainability, doi:10.3390/su14010226_

Round 1
Reviewer 1 Report
The paper offers an interesting approach in the complex structure of continuous quality development of the e-learning process. It focuses on improving algorithms with the ability to extract major themes from textual learning resources, so much needed in everyday mass data production.
However, the lack of guidelines for the structure of the article in the introductory part makes it unnecessarily difficult to follow individual parts of the article. It would be better to shorten the introductory part, or perhaps divide it into two parts, emphasizing the goal of the new algorithm over the existing ones. The introductory part should be shorter and give more consistent guidance to the problem and the article. In it, the reader should also be briefly referred to individual parts of the article, highlighting all the basic key features of their new platform in one place (in a few sentences), describing each individual feature later in the paper. For example, lines 198 – 203 gives an overview of the basic aims of this new e-learning system, but lines 276 – 278 highlight some new important features focused on the self-directed learning process.
Furthermore, the presentation of some results given in the introductory part are sometimes unclear when it comes to the specific reference (for example lines 187 – 190 in connection to the reference [39], one needs to be more specific when using the pronouns “it” or “they” in line 188). When quoting an individual article (throughout the whole article), one should either avoid the use of the pronoun (he, she, they), or use the name of the author (or authors) in connection with a particular article in order to facilitate the monitoring of the structure of the article. (For example, one can use the passive structure, …. In [X] an ensemble method for….. is given…).
Author Response
Response to Reviewer 1 Comments
We are very pleased with your feedback on our research, and we would like to express our deepest thanks and appreciation for your efforts to improve the level of scientific research.
Point 1: The lack of guidelines for the structure of the article in the introductory part makes it unnecessarily difficult to follow individual parts of the article. It would be better to shorten the introductory part, or perhaps divide it into two parts, emphasizing the goal of the new algorithm over the existing ones. The introductory part should be shorter and give more consistent guidance to the problem and the article. In it, the reader should also be briefly referred to individual parts of the article, highlighting all the basic key features of their new platform in one place (in a few sentences), describing each individual feature later in the paper. For example, lines 198 – 203 gives an overview of the basic aims of this new e-learning system, but lines 276 – 278 highlight some new important features focused on the self-directed learning process.
Response 1:
- We have summarized some parts of the introduction and moved the discussion on related work to the new section 2 (Literature Review).
- We have rewritten the last part of the introduction to highlight the key features and contributions of our proposal in one place.
- We have added a last paragraph to the introduction describing the organization of the rest of the paper.
Point 2: The presentation of some results given in the introductory part are sometimes unclear when it comes to the specific reference (for example lines 187 – 190 in connection to the reference [39], one needs to be more specific when using the pronouns “it” or “they” in line 188). When quoting an individual article (throughout the whole article), one should either avoid the use of the pronoun (he, she, they), or use the name of the author (or authors) in connection with a particular article in order to facilitate the monitoring of the structure of the article. (For example, one can use the passive structure, …. In [X] an ensemble method for….. is given…).
Response 2:
- We have revised and reformulated all the references to related work throughout the paper to avoid such Some examples:
- In [16] Arne Holst stated that …
- This algorithm was later improved upon by Federico et al. in [35] …
- Chris et al. used natural language technology …

Reviewer 2 Report
I am honored to have the opportunity to review this study. This study is an interesting study. With the spread of COVID-19, using e-learning is a very feasible solution. I have a few suggestions for the authors.
1) In the introduction part, the innovation of this research needs to be strengthened.
2) There is a lack of literature discussion on learning theory, especially in the part of e-learning, which needs to be supplemented. I also suggest that the author separate the literature review into a subsection.
3) In the methodological part, I find it confusing, I think it should be divided into subsections. For example: What is the main research method? This section looks like an analysis of the website theme, and academic methodology must be added.
4) The result part makes me feel that it is not consistent with the subject of this research. I suggest that the authors adjust the title to fit the subject.
5) For the part that lacks discussion, a discussion section should be added to discuss the results of this study in response to previous studies.
6) There should be research restrictions and practical implications, which must be added at this point.
Lastly, there are some issues with writing and typos which need to be addressed.
Good luck with the revisions.
Author Response
Response to Reviewer 2 Comments
We are very pleased with your feedback on our research, and we would like to express our deepest thanks and appreciation for your efforts to improve the level of scientific research.
Point 1: In the introduction part, the innovation of this research needs to be strengthened.

Response 1:
- We have rewritten the last part of the introduction to highlight the key features and contributions of our proposal in one place.
Point 2: There is a lack of literature discussion on learning theory, especially in the part of e-learning, which needs to be supplemented. I also suggest that the author separate the literature review into a subsection.
Response 2:
- We have moved the discussion on related work from the introduction to the new section 2 (Literature Review).
- We have added a discussion on learning theory to the literature review, and identified the specific learning theory that inspires our contributions.
Point 3: In the methodological part, I find it confusing, I think it should be divided into subsections. For example: What is the main research method? This section looks like an analysis of the website theme, and academic methodology must be added.
Response 3:
- We have added two new paragraphs at the beginning of section 3 (Materials and Methods) to better explain the research method we applied. In them, we summarize again the main contribution of our work and briefly describe how we evaluated it, leaving further details on that for section 4 (Results).
- We have also improved the beginning of section 4 (Results) to summarize the objective of our experiments and the dataset we used.
Point 4: The result part makes me feel that it is not consistent with the subject of this research. I suggest that the authors adjust the title to fit the subject.
Response 4:
- We sincerely think the title of the article is consistent with the research we present. We hope that now we have improved the presentation of our main contributions and the connection with the title is clearer.
- Our main contributions are: an algorithm for extracting the main topics of textual learning resources, and the use of those topics to construct interactive knowledge graphs in which related learning resources are connected so that learners can better explore and browse them. The evaluation we present is focused on testing the accuracy of the topics the algorithm extracts. We believe that the title “Topic Extraction and Interactive Knowledge Graphs for Learning Resources” captures that.
Point 5: For the part that lacks discussion, a discussion section should be added to discuss the results of this study in response to previous studies.
Response 5:
- We have extended section 5 (Discussion) to discuss the results of our study in response to previous research.
Point 6: There should be research restrictions and practical implications, which must be added at this point
Response 6:
- We have extended section 5 (Discussion) to also discuss research restrictions and practical implications of our research.
Point 7: Lastly, there are some issues with writing and typos which need to be addressed.
Response 7:
- We have revised the writing of the paper and fixed some typos and other language issues.

Round 2
Reviewer 1 Report
The manuscript deals with an interesting and important problem of reducing huge amounts of data that make it difficult to strengthen interdisciplinary conceptual links in the learning process. Recognizing the basic concepts of the field and strengthening their interdisciplinary links provides a solid foundation for a qualitative approach to learning, and as the authors interestingly pointed out, this new approach can prevent the development of boredom in the learning process. It makes sense to provide the further development of this algorithm integrating different input data format.Reviewer 2 Report
Regarding the review opinions, the researchers have responded to corrections. There is one point in the content that needs to be adjusted.
1.Format and typesetting should be revised on lines 503 and 504.